# 🔥 STAMO: UNSUPERVISED LEARNING OF GENERAL-IZABLE ROBOT MOTION FROM COMPACT STATE REPRESENTATION

## ABSTRACT

A fundamental challenge in embodied intelligence is developing expressive and compact state representations for efficient world modeling and decision making. However, existing methods often fail to achieve this balance of compactness and expressivity, yielding representations that are either overly redundant or lacking in task-critical information. We propose an unsupervised approach that learns a highly compressed two-token state representation using a lightweight encoder and a pre-trained Diffusion Transformer (DiT) decoder, capitalizing on its strong generative prior. Our representation is efficient, interpretable, and integrates seamlessly into existing VLA-based models, improving performance by **14.3%** on LIBERO and **30%** in real-world task success rate with minimal inference overhead. More importantly, we find that the difference between these tokens, obtained via latent interpolation, naturally represents the motion, which can be further decoded into executable robot actions. This emergent capability reveals that our representation captures dynamics without explicit motion supervision. We name our method *StaMo* for its ability to learn generalizable robotic *Motion* from compact *State* representation, which is encoded from static images, *challenging the prevalent dependence to learning robotic motions with complex temporal modeling and video data.* Our learned representations also enhance policy co-training, outperforming prior methods by **10.4%** with improved interpretability. Moreover, our approach scales effectively across diverse data sources, including real-world robot data, simulation, and human egocentric video.

## 1 INTRODUCTION

*"What we observe as static is merely dynamic equilibrium."*

— Richard Feynman, *The Feynman Lectures on Physics*

Learning reusable and generalizable representations is a cornerstone of intelligent robotics systems. While visual representations in VLAs retain rich perceptual details for multimodal fusion, state representations for world modeling and intermediate reasoning serve a different purpose: they are design to enable efficient future prediction and bridging visual planning with action execution. This role demands two key properties: ***First***, it must be highly compact. Unlike rich visual embeddings used for perception, a state representation's proximity to action generation demands a focus on actionable information to ensure efficient future prediction. ***Second***, it must be expressive despite its compactness. Common low-dimensional approaches, such as trajectories (Wen et al., 2023) or flow fields (Gao et al., 2024; Xu et al.), often fail on this front; they capture basic motion but lack the semantic richness to encode goal states, interaction dynamics, or structured spatial relationships.

Building on these principles, we propose a compact state representation learned using a Diffusion Autoencoder fine-tuned on robotics data. Inspired by prior work (Zhao et al., 2024) that applied Diffusion Autoencoders to facilitate long video generation in VLMs, our method compresses an image into a highly compact latent state (as few as 2 tokens of 1024 dimensions). To ensure the representation is rich, we initialize its decoder from a powerful DiT model pretrained on internet

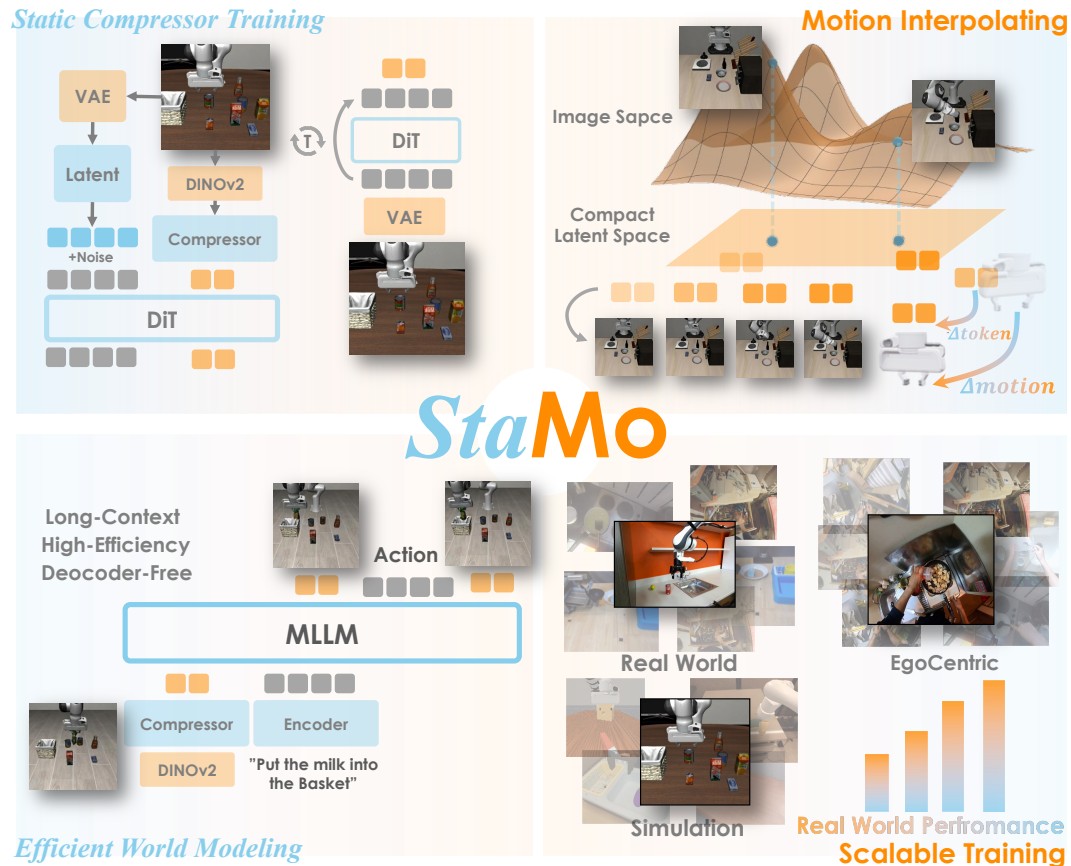

Figure 1: **An overview of our *StaMo* framework.** Our method efficiently compresses and encodes robotic visual representations, enabling the learning of a compact state representation. Motion naturally emerges as the difference between these states in the highly compressed token space. This approach facilitates efficient world modeling and demonstrates strong generalization, with the potential to scale up with more data.

data (Esser et al., 2024), reasoning that the ability to reconstruct pixels accurately requires an implicit understanding of key state information like robot poses and object interactions. This representation integrates seamlessly into existing VLM architectures, extensive experiments show that our method significantly enhances model performance with negligible overhead to inference speed, achieving improvements of + 14.3% in LIBERO and + 30% in real-world deployments.

Interestingly, from this compact state representation, we make a important discovery: robot motion (which is also known as latent action) naturally emerges from the state representation space. By simply performing a linear interpolation between the latent encodings of start and goal observations, we can generate smooth, plausible, and dynamically consistent motion trajectories. This observation provides an alternative to the dominant paradigm of learning latent actions from video data (Ye et al., 2024a; Bu et al., 2025b; Yang et al., 2025; Chen et al., 2024a). While using video seems intuitive as action is inherently temporal, it presents significant challenges: it demands complex, computationally expensive temporal models and often produces ambiguous, coarse-grained actions. This is because high-variance motion within video clips causes models to learn averaged-out representations, making per-frame learning both inefficient and representationally flawed.

These challenges motivated us to revisit the fundamental motivation for using video data, **if the objective is to capture latent actions through frame-wise changes, why should we commit to learning complex motion extractors from suboptimal state representations?** We therefore raise a more fundamental question: **rather than explicitly modeling motion from sequences, can we learn a sufficiently expressive state representation from individual frames such that the simple**

**difference between two states naturally encapsulates a meaningful latent action?** Our work demonstrates this is not only feasible but also highly advantageous. Compared to traditional video-based models, our state-centric approach is more training-efficient and avoids the representational ambiguity caused by motion variations in video. The emergent latent actions exhibit strong generalization and seamless sim-to-real transferability. We validate this effectiveness through extensive co-training experiments, where our latent actions achieve superior performance over traditional and more complex latent action learning approaches.

The captivating properties of state and motion ultimately form the foundation of our method, dubbed *StaMo*, which learns generalizable *Motion* from compact *State representation*. While the state is *static*, the motion is *dynamic*, a harmonious balance is elegantly achieved in our approach. We hope our method would shed new light on future research. Our contributions are summarized as follows:

- We propose *StaMo*, a novel framework that encodes a compact state representation from static images, from which motion naturally emerges.
- Our representation can be efficiently utilized for world modeling and serves as an intermediate representation that bridges vision-language models (VLMs) and action expert modules. It can be seamlessly integrated into existing VLM-based frameworks, delivering improved performance with minimal inference overhead.
- Our motion extraction approach offers enhanced flexibility, strong generalization capability, and excellent transferability. It can be effectively utilized for co-training downstream models and facilitating goal-image planning tasks.
- Comprehensive simulation and real-world experiments, along with extensive visualizations, demonstrate the effectiveness of our approach, which can be readily scaled up with additional data.

## 2 RELATED WORKS

### 2.1 ROBOT-CENTRIC REPRESENTATION LEARNING

Previous robotics research on visual representations has often faced a fundamental trade-off: methods excelling at motion representation, like latent actions (Bu et al., 2025b; Ye et al., 2024a; Cheang et al., 2024b; Bjorck et al., 2025; Gao et al., 2025), flow (Gao et al., 2024; Xu et al., 2024), or trajectories (Wen et al., 2023), typically lack a rich, compact state representation. Conversely, approaches that encode detailed state information from raw images (Wang et al., 2025; Bharadhwaj et al., 2024) or dense features (Zhang et al., 2025; Li et al., 2025a; Nair et al., 2022; Xiao et al., 2022; Majumdar et al., 2023; Radosavovic et al., 2023) are often too high-dimensional and redundant to effectively represent motion via simple differences. *StaMo*, as shown in Figure 2, overcomes this dichotomy by achieving a unique balance. It learns to compress and encode a robot's visual state into a highly compact token space that is both expressive enough for complex tasks and minimal enough that motion can be elegantly and powerfully represented as the difference between two states. This unified approach not only enables efficient world modeling but also demonstrates superior generalization and scalability, providing a robust foundation for future robotic systems.

### 2.2 WORLD MODELING IN VISION-LANGUAGE-ATION MODELS

Benefiting from powerful vision-language models (VLMs) (Bai et al., 2025; Chen et al., 2024b; Beyer et al., 2024) pre-trained on large-scale internet data, vision-language-action (VLA) models have demonstrated significant potential in generalizing across manipulation tasks. Typically, VLAs (Brohan et al., 2022; Zitkovich et al., 2023; Cheang et al., 2024a; Ha et al., 2023; Kim et al., 2024; Black et al.; Wen et al., 2025; Team et al., 2024) map visual information and language instructions into the robot's action space through end-to-end training. A natural extension is to endow the model with world modeling capabilities. Inspired by recent unified generative and comprehension models, a promising direction is to enable large models to reason with images—predicting future visual states while inferring actions or language, thereby creating a mutually reinforcing process. Several works (Wang et al., 2025; Zhang et al., 2025; Cen et al., 2025; Li et al., 2025b) have explored this idea. However, these approaches either require decoding full images during inference or rely on overly redundant state representations, limiting their generalization to novel scenes. In

contrast, our method employs a more compact representation that improves model performance with minimal impact on inference speed. Furthermore, our representation exhibits strong generalization ability, enabling adaptation to unseen scenes without requiring further fine-tuning of the encoder.

### 2.3 Latent Action Learning

Robot learning scalability is constrained by scarce and heterogeneous robotic data, prompting the use of large-scale, action-free internet videos to instill foundational physical and operational knowledge for improved generalization and data efficiency (McCarthy et al., 2024). Numerous studies have explored learning discrete (Ye et al., 2024a; Chen et al., 2024a; Bruce et al., 2024; Bu et al., 2025b; Schmidt & Jiang, 2023; Bu et al., 2025a) or continuous (Yang et al., 2025; Gao et al., 2025) latent actions from unlabeled data and have demonstrated their effectiveness across a wide range of downstream tasks. However, most of these approaches rely on carefully designed model architectures and depend on extracting frames from continuous videos, which introduces limitations such as sensitivity to frame sampling intervals, potential temporal biases, and poorly interpretable, blurry action representations. In contrast, our approach reveals that actionable representations can inherently emerge from large-scale natural image collections—challenging the conventional belief that latent actions must be learned from video sequences. We further demonstrate that such image-emergent actions exhibit superior interpretability and stronger generalization capabilities.

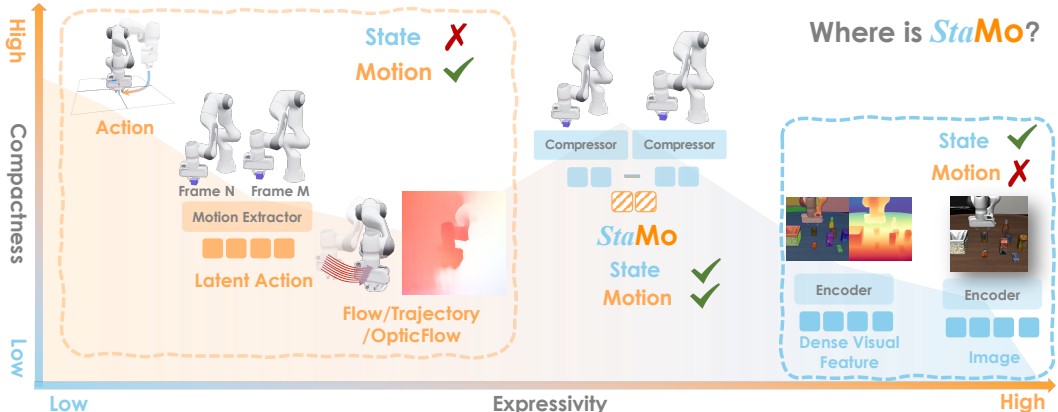

Figure 2: **Where is *StaMo*?** This figure visualizes how different robotic representations fall on the spectrum of expressiveness versus compactness. StaMo uniquely occupies the ideal position, offering both a rich, expressive state representation and the ability to model motion from a highly compact space.

## 3 Method

### 3.1 Compress Images into Compact Representations

A straightforward approach to state representation is to employ pretrained image encoders to process observations, using their output features as latent states. However, this strategy yields prohibitively large feature maps (e.g., $256 \times 1024$), which contain significant redundant information and impose limitations on real-time execution and long-horizon planning in robotics. While using the single [CLS] token from the large feature maps substantially reduces dimensionality, this representation is often too coarse to facilitate effective and precise task execution. To address this trade-off, we draw inspiration from recent work (Zhao et al., 2024) using Diffusion Autoencoders, which have successfully compressed keyframes into compact, expressive latents for video generation in VLMs, hinting at their capability as world models. We adopt a similar strategy and fine-tune a Diffusion Autoencoder specifically for robotic settings, which successfully compresses the observation into a compact latent state of as few as two 1024-dimensional tokens.

To this end, as illustrated in Figure 1, the encoder of our Diffusion Autoencoder, denoted as $\mathcal{E}$, consists of a pretrained DINOv2 model (Oquab et al., 2023) serving as a frozen feature extractor,

which is followed by a transformer-based compressor, to map the input observations into a sequence of compact token representations. The decoder $\mathcal{D}$, is a Diffusion Transformer (DiT) (Peebles & Xie, 2023) conditioned on these tokens to reconstruct the original image observations. Our implementation is built upon the Stable Diffusion 3, where only the transformer-based compressor and the DiT decoder are trained. The DINOv2 encoder remains frozen throughout the training process. We employ the same Flow Matching (Albergo & Vanden-Eijnden, 2022; Liu et al., 2022; Lipman et al., 2022) objective used in base model training to optimize the model, which minimizes the loss between the velocity predicted by $\mathcal{D}$ and the target velocity $u$:

$$
\begin{aligned}
\mathbf{z}_0 &= \tau(\mathbf{x}_0), \\
L_{DAE} &= \mathbb{E}_{\mathbf{z}_0,t}\|\mathcal{D}(\mathbf{z}_t, \mathcal{E}(\mathbf{x}_0), t) - u(\mathbf{z}_t)\|_2^2,
\end{aligned}
\tag{1}
$$

where $\mathbf{x}_0$ is the input image observation and $\tau$ is the VAE encoder of pretrained diffusion model to produce latent $\mathbf{z}_0$. $\mathbf{z}_t = (1 - \sigma_t)\mathbf{z}_0 + \sigma_t\epsilon$ is the linear interpolation between pure noise $\epsilon$ and $\mathbf{z}_0$. After training, The trained encoder $\mathcal{E}$ is able to yield the state representation $s$. Please refer to Appendix A for more details about architecture and training process.

### 3.2 COMPACT REPRESENTATIONS FOR BOTH STATE AND MOTION

A key advantage of our learned representation is its capacity to serve a dual role, representing both state and motion. This duality resolves a fundamental trade-off inherent in conventional robotics representations, which we analyze comparatively in Figure 2. On one hand, traditional motion representations such as end-effector poses, optical flow, trajectories, or recent latent action models, are typically low-dimensional and compact. Their strength lies in capturing dynamics, often through simple differencing, making them effective for representing actions. However, this compactness comes at the cost of expressivity: they lack the rich visual context necessary to reconstruct a plausible or detailed state. On the other hand, prevailing state representations, derived from encoding raw image observations or supplementary inputs like depth and segmentation maps, capture rich semantic and geometric details. While highly expressive, these high-dimensional features are computationally burdensome and fail to intrinsically encode the dynamics or motion required to transition between states. Our approach elegantly bridges this gap. By defining motion as the vector difference between consecutive compact state tokens $a_t = s_{t+1} - s_t$, we achieve a representation that is both compact and expressive, unifying the description of "what the world looks like" and "how the world changes". The effectiveness of representing motions as the differences between states is demonstrated in our experiments in Section 4.4.

### 3.3 STAMO FOR EFFICIENT WORLD MODELING

Existing VLA models, such as our OpenVLA baseline, typically operate as reactive policies. They learn a direct mapping from the current visuolinguistic context to a sequence of low-level actions (e.g., 7-DoF end-effector controls). While effective, this paradigm does not explicitly compel the agent to reason about the physical consequences of its actions or how the world will change as a result. We hypothesize that endowing the agent with a predictive world model to anticipate future visual states will improve the future-planning capability of model. This auxiliary task of predicting "what happens next" should, in turn, regularize the policy and improve the quality of the primary action prediction task.

The compact state representations produced by our StaMo encoder $\mathcal{E}$ are ideally suited for this purpose, which provide a concise yet rich summary of the environment for the model to reason about the future. To this end, we integrate the compact representations of StaMo into the OpenVLA architecture and train the model to jointly predict the next state and the corresponding action. Technically, we achieve this by attaching a lightweight MLP head to OpenVLA's autoregressive backbone, which is tasked specifically with predicting the subsequent state representation. The model is optimized with a composite loss function that balances action generation and future-state prediction:

$$
L_{total} = \lambda_{action}L_{action} + \lambda_{future}(L_{mse}(s_{pred}, s_{gt}) + L_1(s_{pred}, s_{gt})),
\tag{2}
$$

where $L_{action}$ is the standard cross-entropy loss for next-token prediction, anchoring the model to its primary control task. The world model objective combines Mean Squared Error (MSE) and L1

losses to enforce accurate regression of the ground-truth future state. In our experiments, we set $\lambda_{action} = \lambda_{future}$, reflecting our hypothesis that learning to predict is as crucial as learning to act. Our experimental results validate this hypothesis, demonstrating that compelling the VLA to predict future states significantly improves task success rates, as detailed in Section 4.2.

### 3.4 STAMO FOR LATENT MOTIONS CO-TRAINING

While Section 4.1 provides a qualitative visualization of the emergent motion, the abstract nature of our latent motion representation makes direct quantitative comparison challenging. To rigorously assess its effectiveness, we employ a policy co-training strategy. This approach evaluates the utility of our motion representation on downstream tasks by jointly training a policy model on a small set of action-labeled robot data alongside a larger set of action-less video data. We infer a latent motion, $m_t$, for each pair of consecutive video frames by calculating the difference between their compact state representations encoded by our model $E(\cdot) : m_t = E(o_{t+1}) - E(o_t)$, This process generates pseudo-action labels for the video data, creating a unified dataset where our emergent motions and ground-truth robot actions are learned jointly within a single policy model. This approach unlocks the potential to learn from vast and diverse video sources without requiring explicit action labels.

We tested this co-training framework with both in-domain robot videos and out-of-domain human demonstration videos to assess the generalization of our motion representation. As shown in Section 4.3, this method leads to substantial performance gains compared with previous method. The results confirm that the motions emerging from *StaMo* serve as an effective and robust proxy for true actions, providing a scalable pathway to enhance policy learning by leveraging the wealth of unlabeled visual data readily available.

## 4 EXPERIMENTS

### 4.1 QUALITATIVE ANALYSIS OF STAMO

In this subsection, we present a qualitative visualization and analysis of the *StaMo* results. *StaMo* demonstrates the ability to perform high-quality reconstructions of robotic manipulation images with strong generalization capabilities. It achieves excellent results on both in-domain and out-of-domain data. Quantitative reconstruction results are shown in Table 1, and some qualitative results are presented in Figure 3.

Figure 3 also showcases *StaMo*'s powerful motion interpolation ability. By linearly interpolating the tokens in the latent space, the decoded images exhibit excellent continuity and plausible motion. Furthermore, we demonstrate the results of motion transfer. By taking the difference between tokens in the latent space, the resulting latent motion proves to be highly transferable—both in sim-to-sim, sim-to-real, and real-to-sim scenarios. This indicates that the learned motion is not scene-specific but possesses strong generalization capabilities.

Table 1: Reconstruct Performance comparison of different datasets using our *StaMo* encoder.

| Dataset | libero_10 | libero_90 | libero_goal | libero_object | libero_spatial | Droid | Maniskill(OOD) |
|---|---|---|---|---|---|---|---|
| PSNR (dB) | 25.5194 | 27.2470 | 24.6467 | 27.0011 | 25.9683 | 20.2492 | 22.1673 |
| SSIM | 0.8909 | 0.8962 | 0.8926 | 0.9105 | 0.8984 | 0.7346 | 0.8824 |

### 4.2 WORLD MODELING RESULTS

In this section, we evaluate the application of *StaMo* for efficient world modeling. A key advantage of our approach is its ability to integrate seamlessly into existing VLM-based frameworks. We demonstrate this by conducting experiments on two strong baselines: OpenVLA and its successor, OpenVLA-OFT. We specifically chose OpenVLA-OFT because it introduces architectural improvements like parallel decoding and action chunking,

Table 2: Inference frequency.

| Method | Frequency (Hz) |
|---|---|
| UniVLA | 2.65 |
| OpenVLA | 4.16 |
| OpenVLA + *StaMo* | 4.02 |

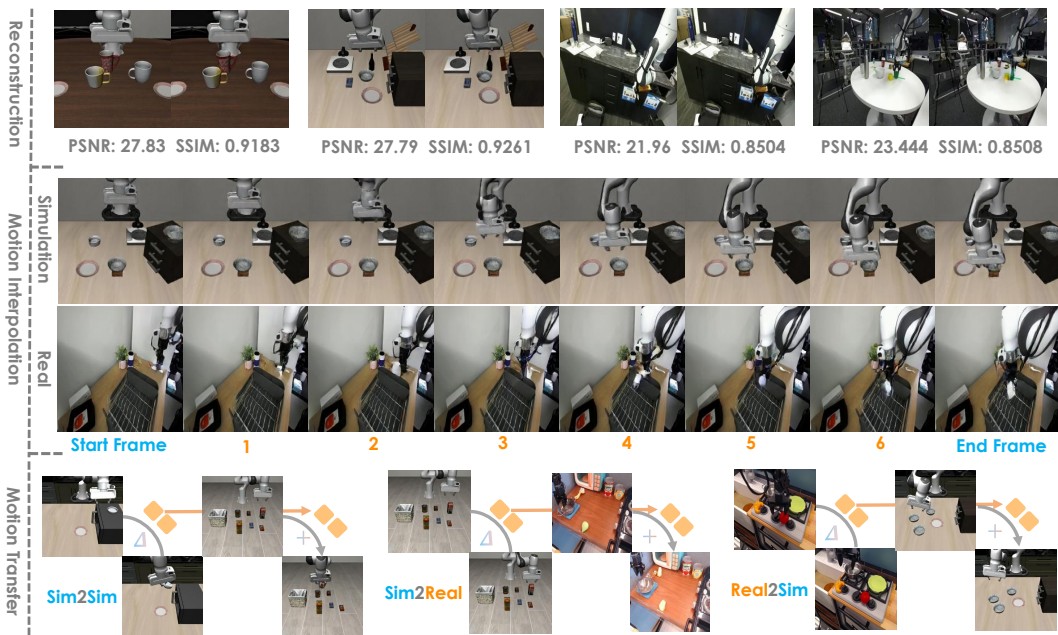

Figure 3: **Reconstruct images using our StaMo encoder with as few as two 1024-dimensional tokens.** The first row shows the ground truth, and the second row shows the predicted results, with corresponding PSNR and SSIM metrics listed below. The results demonstrate that StaMo can preserve high image fidelity and structural similarity even under extremely compressed state representations.

which result in a significantly longer action horizon. This requires the model to predict a state representation at a more distant future step, making it a more challenging and relevant benchmark for world modeling.

Both the StaMo state and motion representations can serve as predictive targets for a world model, and their performance is detailed in Table 3. Our findings indicate that the optimal choice of representation depends on the policy's action horizon. For the standard OpenVLA, which predicts actions over a short, single-step interval, the motion representation yields better performance. This is because the latent motion is conceptually analogous to the delta end-effector (EE) pose used in frameworks like LIBERO, providing a direct and incremental target. Conversely, for OpenVLA-OFT, which plans over a longer execution step, the state representation is recommended, as it acts as a stable goal-conditioning signal. Importantly, as we show in Table 2, the integration of StaMo introduces negligible overhead, leaving the model's inference speed almost entirely unaffected.

Table 3: Performance comparison of different methods (%).

| Method | Spatial | Object | Goal | Long | Average |
|---|---|---|---|---|---|
| OpenVLA | 80.2 | 81.3 | 75.8 | 49.7 | 71.7 |
| OpenVLA + StaMo state | 91.4 | 92.3 | **87.1** | 73.2 | 86 |
| OpenVLA + StaMo motion | **92.3** | **92.5** | 86.4 | **75.1** | **86.6** |
| OpenVLA-OFT* | 91.7 | 93.2 | 89.6 | 90.3 | 91.2 |
| OpenVLA-OFT* + StaMo state | **94.8** | **96.2** | **92.2** | **92.9** | **94.0** |
| OpenVLA-OFT* + StaMo motion | 92.3 | 94.1 | 90.2 | 92.1 | 92.1 |

Table 4: Performance comparison of RDT with different data configurations (%).

| Method | Spatial | Object | Goal | Long | Average |
|---|---|---|---|---|---|
| RDT (all Real) | 91.6 | 93.3 | 86.7 | 73.3 | 86.2 |
| RDT (1 Real) | 71.7 | 70.0 | 66.7 | 43.3 | 62.9 |
| RDT (1Real+ 4ATM) | 83.3 | 81.7 | 71.7 | 56.7 | 73.4 |
| RDT (1Real+ 4LAPA) | 80.0 | 76.7 | 75.0 | 65.0 | 74.2 |
| RDT (1Real + 4StaMo) | **90.0** | **91.6** | **86.7** | **70.0** | **84.6** |

## 4.3 POLICY CO-TRAINING RESULTS

In Table 4, we detail the results of our policy co-training. Our approach utilizes a DIT-based model on the RDT architecture, trained on a per-task dataset of 10 robot trajectories and 40 video trajectories with StaMo-generated pseudo-actions. We evaluated the final model checkpoint for each task across 20 trials, averaging the success rate over three full runs. StaMo significantly outperforms both LAPA and ATM, suggesting that its pseudo-action labels are more faithful to the original actions.

## 4.4 ACTION LINEAR PROBING EXPERIMENT

A central claim of our work is that an effective latent action can be formulated as the simple vector difference between two latent states encoded by *StaMo*. To quantitatively validate the quality and utility of these emergent latent actions, we employ a linear probing protocol (Alain & Bengio, 2016), a standard methodology for evaluating the efficacy of learned representations, to measure how much explicit, predictable information about the motion is contained within our latent action formulation.

We first construct a dataset by randomly sampling tuples from a large collection of robot trajectories. Each sampled tuple is of the form $(\mathbf{I}_n, \mathbf{I}_{n+k}, \mathbf{A}_n)$, where $\mathbf{I}_n$ is a starting image, $\mathbf{I}_{n+k}$ is the goal image after a horizon of $k$ steps, and $\mathbf{A}_n = (\mathbf{a}_n, \ldots, \mathbf{a}_{n+k-1})$ is the sequence of ground-truth robot actions executed between them. Each image pair is then encoder into latent action, $\Delta z$, using our frozen *StaMo* encoder, $\mathcal{E}$:

$$\Delta z = \mathcal{E}\left(\mathbf{I}_{n+k}\right) - \mathcal{E}\left(\mathbf{I}_n\right). \tag{3}$$

We then train a lightweight Multi-Layer Perceptron (MLP) to predict the action sequence $\mathbf{A}_n$ from the latent action representation $\Delta z$. The performance is measured by the Mean Squared Error (MSE) between the predicted action sequence $\hat{\mathbf{A}}_n$ and the ground-truth sequence $\mathbf{A}_n$. This process can be formally described as follows:

$$\mathrm{MSE}\left(\mathbf{A}_n, \hat{\mathbf{A}}_n\right) = \mathrm{MSE}\left(\mathbf{A}_n, \mathrm{MLP}(\Delta z)\right). \tag{4}$$

A low LP-MSE score provides strong quantitative evidence that the vector difference between our latent states serves as a highly informative and linearly separable representation for robotic motions.

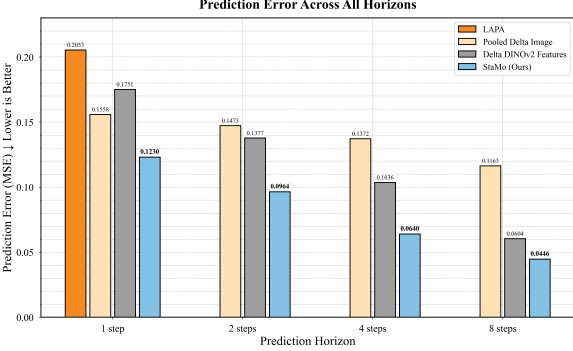
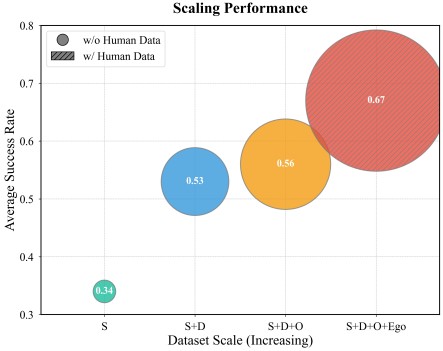

Figure 4: **Linear Probing MSE results.** We compare our method against three baselines. Our method consistently achieves the **lowest** MSE across all horizons.

Figure 5: **Scaling Performance.** The Performace of our model can be steadly scaling with more data, including human egocentric data.

To validate our approach, we benchmark our latent action representation against several baselines. The first set of baselines tests the importance of our structured latent space against alternatives operating on raw data: Pooled Delta Image (using pixel-wise differences) and Delta DINOv2 Features (using feature-space differences). The second baseline, LAPA (Ye et al., 2024b), contrasts our deterministic state-difference formulation with a state-of-the-art autoregressive model that generates latent action tokens.

For a fair comparison, all representations are used to train an identical lightweight MLP. We evaluate non-generative methods across 1, 2, 4, and 8-step prediction horizons, while LAPA is assessed at its native 1-step horizon. Once trained, the MLP can directly map a current observation and a goal image to an executable action trajectory on the test set. The results in Figure 4 confirm the superiority of our method. Additional details are provided in Appendix B.

## 4.5 Real World Experiments

**Experiment Setup.** Our experimental benchmark comprises six tasks—three short-horizon and three long-horizon—to comprehensively evaluate the effectiveness of our representation in facilitating world modeling and decision-making across a spectrum of task complexities, as shown in Figure 6. For each task, we collect 50 demonstrations and evaluate performance over 20 trials, results are shown in Table 5, our real-world robot setting and tasks definition are detailed in Appendix C.

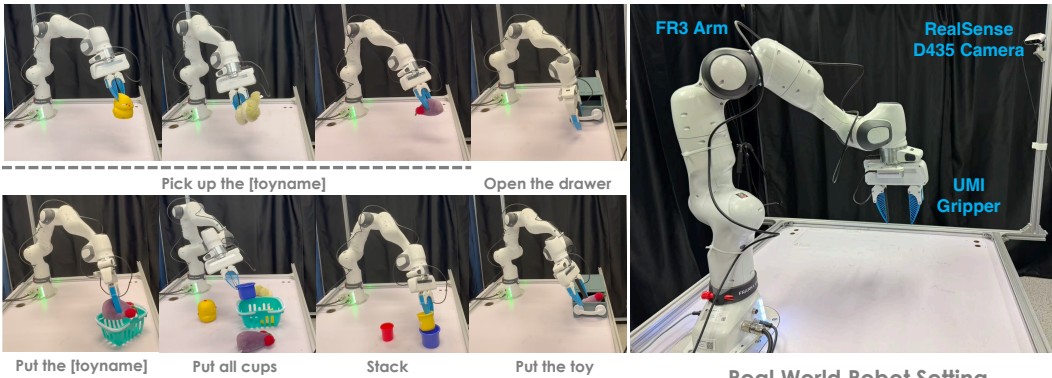

Figure 6: **Real World Setting and Tasks.** We designed a benchmark of six real-world robotics tasks, spanning both short- and long-horizon challenges, to evaluate the effectiveness of our representation for learning world models.

Table 5: Performance comparison on short and long horizon tasks.

| Method | Short horizon tasks | | | | Long horizon tasks | | | | |
| | Pick up the [toyname] | put the toy into the basket | Open the drawer | Average | put all cups into the basket | put the toy into the drawer | Stack all cups | Average | All Average |
| --- | --- | --- | --- | --- | --- | --- | --- | --- | --- |
| OpenVLA | 0.35 | 0.3 | 0.25 | 0.30 | 0.2 | 0.3 | 0.15 | 0.2 | 0.25 |
| OpenVLA + StaMo states(S) | 0.5 | 0.45 | 0.3 | 0.42 | 0.25 | 0.3 | 0.2 | 0.25 | 0.34 |
| OpenVLA + StaMo states(S+D) | 0.6 | 0.6 | 0.55 | 0.58 | 0.45 | 0.6 | 0.45 | 0.5 | 0.53 |
| OpenVLA + StaMo states(S+D+O) | 0.65 | 0.65 | 0.5 | 0.6 | 5 | 0.5 | 0.45 | 0.52 | 5.56 |
| OpenVLA + StaMo states(S+D+O+Ego) | 0.7 | 0.75 | 0.6 | 0.72 | 0.65 | 0.65 | 0.55 | 0.62 | 0.67 |

## 4.6 Scalable Simulation and Real World Performance

A natural question following our initial validation is the scalability of our approach. To explore this, we scaled our training with a diverse corpus of data, including more simulation, the Open X-Embodiment dataset(which contains more different robot embodiments), and egocentric human demonstrations. The results in Table 5 reveal steady performance improvements across both simulated and real-world robotics benchmarks, which clearly depicted in Figure 5, confirming that our model is highly scalable and benefits from diverse data sources.

## 5 Conclusion

In this work, we introduced *StaMo*, a framework that extracts reusable latent action vectors from static images via a transformer-based tokenizer and a diffusion decoder. We demonstrate that linear interpolation in this latent space produces smooth, physically plausible motions, and that these vectors generalize zero-shot to unseen robots—suggesting that large-scale visual models implicitly learn a linearized dynamics manifold. Our approach offers a pathway toward scalable unsupervised skill discovery from diverse visual data, bridging the gap between static perception and dynamic action.

## ETHIC STATEMENT

This research adheres to widely recognized ethical guidelines in robotics and artificial intelligence. All experiments were conducted using publicly available datasets and simulated environments, without involving personal data, or sensitive information. The methods proposed in this work are intended solely for academic research and technological advancement, and do not pose immediate risks of misuse. We encourage the responsible deployment of robotic technologies, ensuring that their applications align with societal values and ethical standards.

## REPODUCIBILITY STATEMENT

To ensure reproducibility, we provide detailed descriptions of the model architectures, training procedures, and evaluation protocols in the Section 3. All experiments were conducted using publicly available datasets (e.g., LIBERO and DROID) and standard simulation environments. Hyperparameters and implementation details are fully documented in Appendix A. The code, configuration files, and instructions necessary to reproduce our results will be released upon publication.

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

USE OF LLMS

We used large language models (LLMs) only for minor assistance in polishing the language and adjusting the presentation of tables. No LLMs were involved in designing the methodology, conducting experiments, or analyzing results.

# A  IMPLEMENTATION DETAILS

Our *StaMo* encoder experiments were conducted on Ubuntu 22.04 LTS with PyTorch 2.1, using publicly available pretrained weights such as DinoV2 and Stable Diffusion 3 for model initialization. Unless otherwise noted, all main experiments were carried out on standard public datasets under a consistent hardware and software setup. Training was performed with the AdamW optimizer, an initial learning rate of 3e-5 with a cosine decay scheduler, a batch size of 512 per GPU, and a weight decay of 1e-3. For evaluation, PSNR and SSIM were adopted as the primary metrics to assess the quality of image reconstruction. All experiments were run on 8 NVIDIA H100 GPUs, with a typical training duration of around ten days, and we fixed the random seed to 33 to ensure reproducibility.

Unless otherwise specified, all main experiments are conducted using data from publicly available datasets, LIBERO and DROID. We strictly adhere to their original release protocols and only perform essential preprocessing steps, without involving any private or sensitive information.

Table 6: Reconstruction performance with **2 tokens** under different hidden dimensions using our *StaMo* encoder.

| hidden_dim | Datasets | libero_10 | libero_90 | libero_goal | libero_object | libero_spatial |
|---|---|---|---|---|---|---|
| 256 | PSNR (dB) | 26.5969 | 28.5858 | **27.5070** | **31.2779** | 25.7993 |
|  | SSIM | 0.9082 | 0.8948 | **0.9327** | 0.9478 | 0.9118 |
| 512 | PSNR (dB) | **27.3322** | **29.5773** | 26.2000 | 31.0129 | 24.6279 |
|  | SSIM | 0.9180 | **0.9127** | 0.9194 | 0.9444 | 0.9009 |
| 1024 | PSNR (dB) | 27.0418 | 29.4810 | 26.4807 | 31.2653 | **25.9964** |
|  | SSIM | **0.9186** | 0.9124 | 0.9194 | **0.9480** | **0.9155** |

As shown in Table 6, all results are obtained from models trained on the libero datasets, where the latent state is represented using only 2 tokens. Across different hidden dimensions (256/512/1024), the model achieves comparable PSNR and SSIM scores on all datasets, with only marginal variations. This indicates that the choice of hidden dimension has limited impact on reconstruction performance, and the model maintains stable effectiveness across a wide range of dimensional settings.

# B  ACTION LINEAR PROBING DETAILS

Our linear probing dataset is constructed by sampling from the LIBERO dataset. We performed a randomized sampling process across all available trajectories to create a dataset of tuples $(\mathbf{I}_n, \mathbf{I}_{n+k}, \mathbf{A}_n)$, as described in the main paper. A total of approximately 20000 samples are collected, which are then partitioned into training and validation sets with a 95%/5% split, respectively. This process is repeated independently for each of our four evaluated action horizons, where $k \in \{1, 2, 4, 8\}$. To ensure a fair comparison, the same random seed are used for data sampling and splitting. Notably, to accommodate the LAPA baseline which requires task description input, each sampled tuple also includes the corresponding language instruction. For the $k = 1$ comparison, the exact same initial frames $(\mathbf{I}_n)$ are maintained through using same random seed across all methods.

Following the standard linear probing protocol, we use a simple Multi-Layer Perceptron (MLP) as the probe to decode the action representations. The MLP consists of a single hidden layer with 128 units and a ReLU activation function. The architecture is intentionally kept minimal to ensure that the measured performance primarily reflects the quality of the input representation rather than the capacity of the probe itself. During training, the upstream encoders (i.e., *StaMo* and DINOv2) are kept frozen, and gradients are only computed and backpropagated for the MLP's parameters.

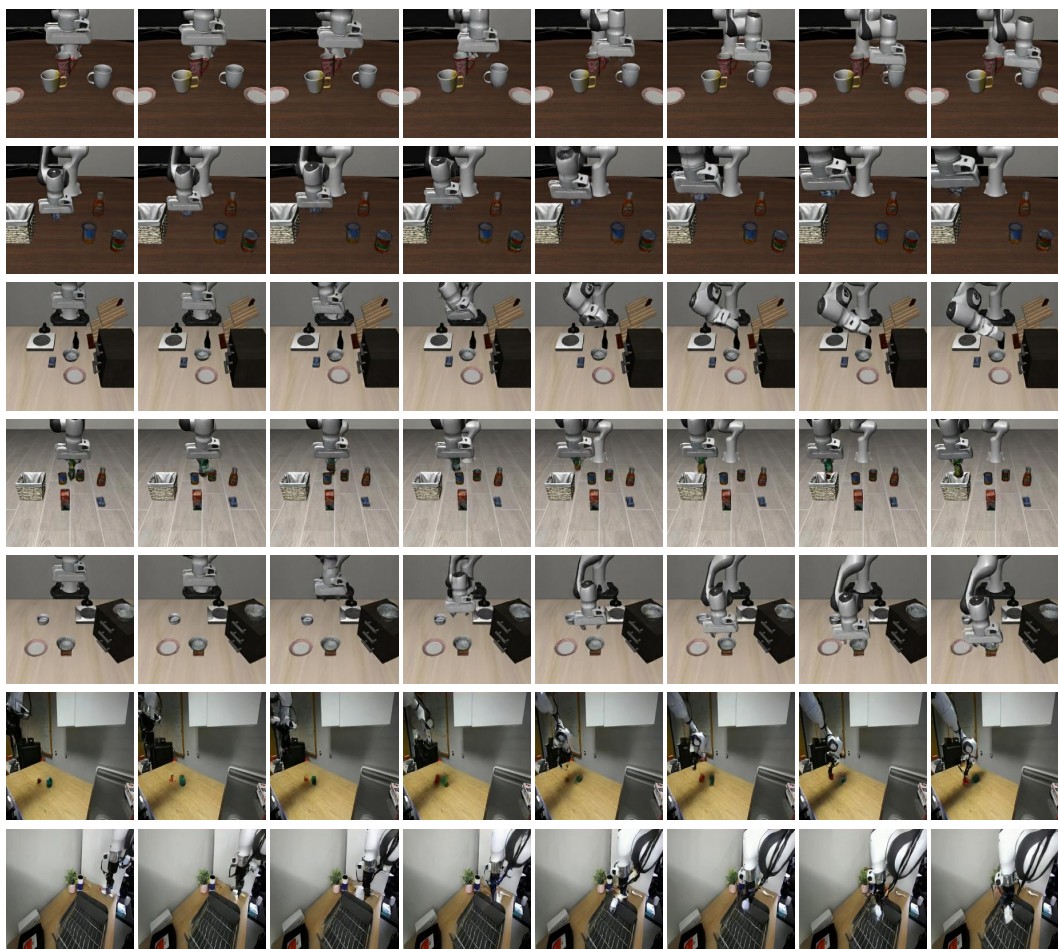

Figure 7: **Visual reconstruction results from the same episode.** The first and last frames are reconstructed from ground-truth images using the *StaMo* encoder, while the intermediate frames are generated by linearly interpolating between the latent state tokens of the two endpoints. The transitions show that both the robotic arm and the objects move in a continuous and smooth manner.

To visually substantiate the rich motion information encoded within our latent actions, we conduct a qualitative replay experiment in the LIBERO simulation environment. For each of the ten tasks in the LIBERO-10 benchmark, we utilize all the ground-truth initial frames ($\mathbf{I}_n$) and final frames ($\mathbf{I}_{n+k}$) from the demonstrations. These two frames are encoded into a single latent actions ($\Delta z$) using our frozen *StaMo* encoder. Here we use $k = 8$ in our experiment.

The latent actions are then fed into the corresponding pretrained MLP probe to predict the entire sequence of $k$ actions. To lower the accumulation of small prediction errors, we only execute the first action in every sequences. The results, visualized in Figure 9, show that even the actions decoded by this simple MLP are sufficient to successfully execute the corresponding tasks to some degree. We also observe that the primary failure mode occurred in tasks requiring precise grasping, where the predicted actions sometimes cause the gripper to descend to an insufficient depth.

## C REAL WORLD SETTING

Our robotic setup includes a single Franka Research3 arm equipped with a UMI (Chi et al., 2024) grippper. A third-person-view RealSense D435 camera is mounted in a fixed position to capture environmental observations at a resolution of 640×480 pixels. Following the implementation in

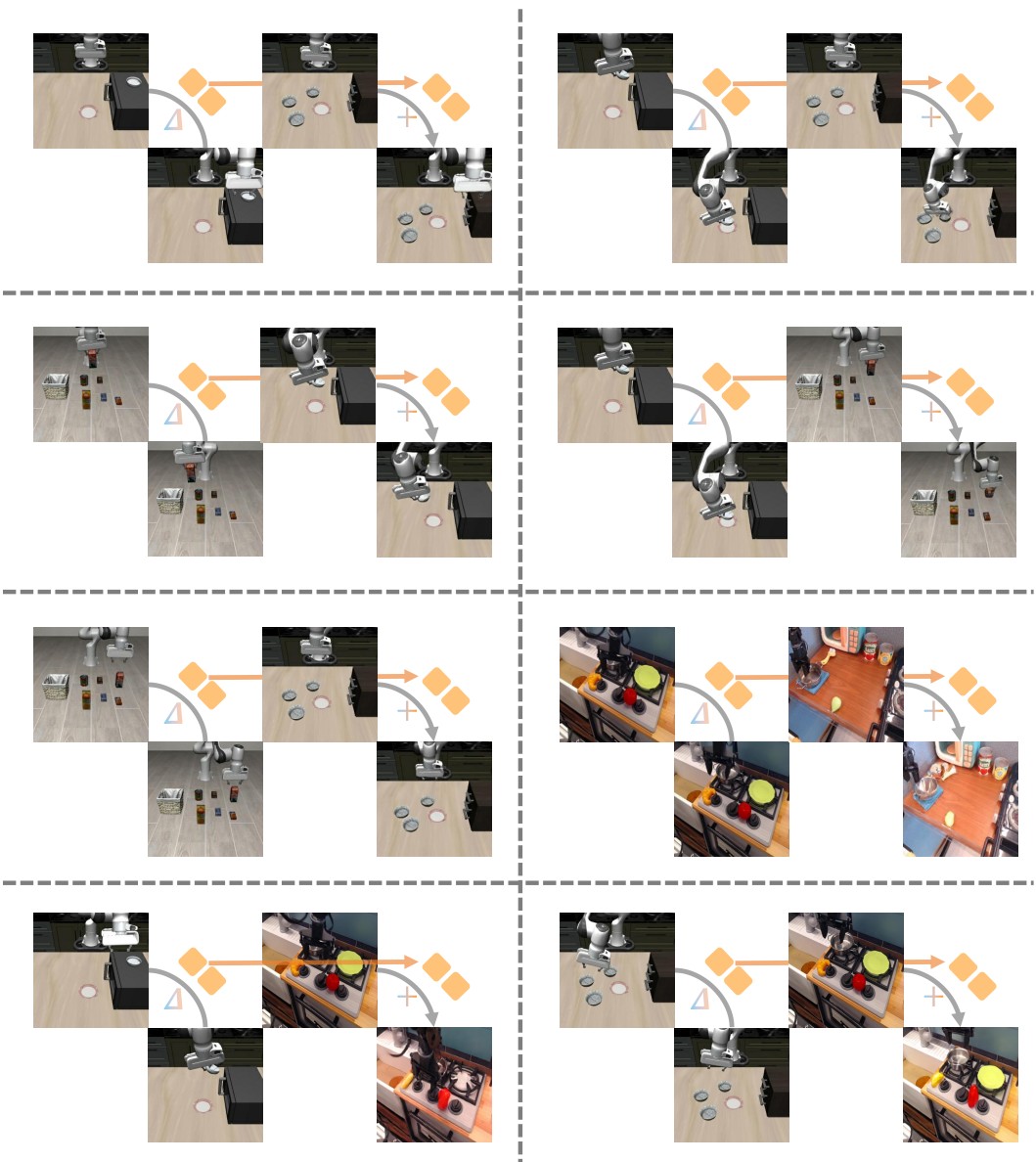

Figure 8: **Transfer linear interpolation experiment with the *StaMo* encoder.** The left and right panels illustrate different task scenarios, where reconstructions are obtained by tokens(3) + tokens(2) − tokens(1), demonstrating the linear interpolation property of latent representations during transfer.

publicly available code [1], we use a 3D mouse for teleoperated data collection. Our system operates at 20 Hz, which is a moderate reduction from the native 100 Hz control frequency to balance training efficiency and motion continuity. Actions are defined as absolute end effector poses in SE(3), using 3D position and quaternion orientation. Our task definition are list below:

**1) Pick up the [TOY NAME]:** The robot is required to pick up the specific toy, including the toy snake, the toy eggplant, and the toy chicken, resulting in a total of three tasks (with no distractions).

**2) Put the toy into the basket:** The robot need to pick up the toy(no other distractions) and put it into the basket.

**3) Open the drawer:** The robot is required to approach the drawer and pull it open.

---

[1]https://github.com/UT-Austin-RPL/deoxys_control

Figure 9: **MLP-predicted actions replay.** Actions are predicted by an MLP probe from our *StaMo* latent action representation and replayed in LIBERO-10 tasks. The results show that our latent actions are remarkably effective, encoding functionally coherent and executable motions.

**4) Put all cups into the basket:** We place two cups with different colors on the table and set a few other things(toys) as distractions. The robot need to sequentially put two cups into the basket.

**5) Put the toy into the drawer:** We place a toy on the table and set a few other things as distractions. The robot need to sequentially open the drawer, pick up the specific toy and put it into the drawer, and close the drawer.

**6) Stack all cups:** Three cups of varying sizes are placed on a tabletop. The robot's task is to perceive their relative sizes and then stack them in descending order (from largest to smallest).drawer, and close the drawer.

