# OpenReview forum: "StaMo: Unsupervised Learning of Generalizable Robot Motion from Compact State Representation"
_ICLR.cc/2026/Conference — ICLR 2026 Conference Withdrawn Submission_

### Official Review · Reviewer_cJUi · 2025-10-24

**Soundness:** 3
**Presentation:** 3
**Contribution:** 2
**Rating:** 4
**Confidence:** 4

**Summary:**

The authors present a way of extracting "latent" states from images in VLA training, analogous to LAPA. The idea is to simply encode the "state" of an image as tokens from a diffusion auto-encoder trained via flow-matching. Importantly, parts of the auto-encoder are initialized from pretrained models like Dino and Stable Diffusion. These "latent states" can be used as labels for adding a world modeling loss to a base VLM, which improves performance as per Table 3. It can also be used for "co-training" on video-only data, which improves performance as per Table 4. The authors show that these gains exist in the real world (Table 5) and provide some qualitative and linear probing experiments on the representations.

**Strengths:**

- The idea is simple and well motivated
- The authors do multiple evaluations in real and sim
- The paper is very well written

**Weaknesses:**

- The novelty is limited
- Comparison to baseline methods like Flare is missing. Does the proposed method beat Flare in e.g,. Table 3?
- It is not clear why state compactness is desirable.

**Questions:**

1. Fig 1 says "Image Sapce", please fix this.
2. How come the inference speed is unaffected in table 2? A whole forward pass of the image encoder is needed right?
3. Why are you optimizing for state-compactness? Isn't inference speed a better benchmark to target?
4. Why is both L1 and L2 loss used in equation 2?
5. Could you try a stronger base VLA than openVLA?
6. In table 5, for the second to last row, the value is ~10x larger than other rows. Is this a typo?

---

### Official Review · Reviewer_94H6 · 2025-11-01

**Soundness:** 2
**Presentation:** 3
**Contribution:** 1
**Rating:** 2
**Confidence:** 3

**Summary:**

This paper proposes an unsupervised diffusion-autoencoder that compresses a single image observation into an extremely compact two-token state (each 1024-D), using a frozen DINOv2 encoder with a lightweight transformer “compressor” and a DiT decoder trained with flow-matching. The core claim is that robot motion emerges as the simple difference of two states. The authors use StaMo states and motions as targets for world-modeling heads on VLA policies, perform policy co-training by generating pseudo-actions from unlabeled videos via state differences, and report both simulation and real-robot gains with low inference overhead. Experimental results validates the effectiveness of the proposed method in simulated and real-world tasks.

**Strengths:**

* Good visibility of figures
* Easy to read
* This paper provide evaluation results both on simulated and real-world robotics benchmark

**Weaknesses:**

* Major concern on its limited novelty
    * While the paper is written well, the proposed method seems to barely have their unique contributions.
        * This paper slightly contributed beyond applying Zhao et al. [1]. Specifically, I understand StaMo as a naive extension of the prior work (Zhao et al. [1]) to the robot tasks. Lines 209-214 also indicate that StaMo directly employ Zhao et al.'s method, with slight architectural changes, as described in Lines 215-218. Consequently, this seems more like an evaluation of prior work into new benchmark in robot tasks. While I also agree that this paper contributed in terms of evaluation, its contribution in substantive methodological or conceptual improvements is weak
        * The above aspect is further revealed in the following elements:
            * Among four subsections in the Method section, only Section 3.1 and 3.2 cover methodological explanation and detail of StaMo itself, which do not exceed even 1 page. Section 3.3 and 3.4 covers about how to adapt StaMo to robotics tasks
            * Section 3.1 explains that StaMo employ Zhao et al. with Dinov2 and DiT, and adopt flow matching loss that commonly used in the previous works
            * Moreover, even in the explanation in Section 3.1 and 3.2, most sentences only described existing studies rather than new mechanisms.
                * The core claim in Section 3.2 is tha a_t = s_{t+1} - s_t. However, this simple claim is expanded into 16 lines. Given that "feature gap between two consecutive frames can be used as motion features" is widely understood that it can be employed without long supportive explanation, this lengthy explanation heightens concerns about limited originality

* Clarity improvement in Figure 1
    * gray, orange, blue tokens should be described at least in caption or figure.
* Minor issues
    * Erratum in Figure 1: "Deocoder-Free"

[1] Canyu Zhao, Mingyu Liu, Wen Wang, Weihua Chen, Fan Wang, Hao Chen, Bo Zhang, and Chunhua Shen. Moviedreamer: Hierarchical generation for coherent long visual sequence. arXiv preprint arXiv:2407.16655, 2024.

**Questions:**

* How did you compare the compactness and expressivity of the described methods in Figure 2?

---

### Official Review · Reviewer_FTP6 · 2025-11-02

**Soundness:** 1
**Presentation:** 1
**Contribution:** 2
**Rating:** 2
**Confidence:** 4

**Summary:**

This paper introduces StaMo, a compressed visual representation for robotic policies. StaMo compresses DINOv2 patch tokens to two tokens per frame with a diffusion autoencoder for more compact state representation, and uses representation deltas as the latent action. The compressed representation works for policy and world model learning. Experiments on LIBERO and real experiments shows that StaMo representations improves world modeling over vanilla OpenVLA, and cotraining on StaMo latents improves policy performance over prior state of the art.

**Strengths:**

- Compressing DINOv2 tokens to a compact representation and using deltas as latent action is well-motivated.
- StaMo representation shows improvement for world modeling in LIBERO with negligible inference slowdown with OpenVLA.
- StaMo latent cotraining improves over LAPA/ATM.
- A real robot environment shows StaMo representation improves over baseline RDT.

**Weaknesses:**

- Lack of relevant baselines.
  - The world modeling section only compares against vanilla OpenVLA with an added world modeling object and does not compare with existing world modeling work such as DINO-WM, AdaWorld, etc. Table 1 only contains reconstruction quality metrics for StaMo and no baselines at all. Similarly, in Table 2, the reviewer is not sure what UniVLA is supposed to do here, since it is not used in any experiments below; and OpenVLA + StaMo is slower than vanilla OpenVLA, which undercuts the claim that StaMo enables efficient world modeling. The authors also state that training runs on 8xH100 for multiple days.
  - Unclear what ATM is in the cotraining section.
  - The real experiment section is a single embodiment behavioral cloning setup. No cross-embodiment experiments to support the claims of generalizability and scalability to unseen robots.
- No ablation experiments. Only a hidden dim sweep table in Appendix A.
- The writing could be a bit clearer. I have asked for crucial clarifications in the question section below.

**Questions:**

- A core claim is that StaMo representation is good for efficient world modeling, which is not adequately supported by the limited experiments. What is the inference speed and quality difference between using raw DINOv2 tokens vs. StaMo compressed? How does it compare with established world modeling approaches?
- In the real robot experiments, is the plain OpenVLA fine-tuned on the same dataset? How well does vanilla behavioral cloning (e.g. diffusion policy) do on this dataset, with a simple ImageNet-pretrained ResNet, and with StaMo encoder?
- Could you clarify what ATM is in 4.3?
- What are S, D, O, and Ego in Table 5? And on the fourth row "long horizon tasks", I am assuming that these are success rates, and it seems like the 5 is supposed to be 0.5?

---

### Official Review · Reviewer_WdQt · 2025-11-03

**Soundness:** 3
**Presentation:** 3
**Contribution:** 2
**Rating:** 4
**Confidence:** 4

**Summary:**

This work aim at finding a compact and expressive state representation for robotics. The key design is to leverage a pre-trained vision encoder, and a pre-trained image diffusion model as decoder. To obtain a compact representation, the latent is compressed into 2 tokens/image. Experiments are conducted on both simulation and real-world scenarios. By additionally predicting the learned state, performance is improved in robotic setting.

**Strengths:**

The writing is clear and easy to follow.

Idea is simple but effective.

Experiments sound and comprehensive, revealing some interesting things like interpolation and linear probing.

**Weaknesses:**

Some key details are missed in the main paper. For example, it should specify which data you used for different settings. Also, some metrics are missed in the main paper, e.g., Table 3 & 4, are these success rates? The main paper should be self-contained, and complete experimental setting would make the number valid.

Although low compression rate is achieved, I am worried about some motion are missed in the reconstruction one. For example, in the up-right corner of the Fig 3, I can't tell which action is given the reconstruction result.

Some previous works have studied similar insight. For example, as listed in section 2.2: Wang et al. 2025, Zhang et al. 2025, Cen et al., 2025. Li et al. 2025b. The author claim that these works have limited generation across scenes. This claim needs evidence and it is suggested to provide the comparison with the most related works. Lack of the comparison makes the results not convincing.

Typos: title of section 2.2.

**Questions:**

Refer to the weakness.

---

### Author Response · Authors · 2025-11-14

We would like to express our sincere gratitude to the four reviewers for carefully reading our manuscript and providing valuable comments and suggestions. After thorough consideration, we have decided to withdraw the current version of the paper in order to further polish it. We plan to supplement additional experiments and reformulate the paper to address potential misunderstandings and improve its overall quality. Thank you again for your time and insightful feedback！

---

### Note · Authors · 2025-11-14

I have read and agree with the venue's withdrawal policy on behalf of myself and my co-authors.